# Nonalcoholic fatty liver disease and cognitive impairment: A prospective cohort study

**Mary Cushman** [1,2]*, **Peter W. Callas**[1,3], **Kristine S. Alexander**[1], **Virginia Wadley**[4], **Neil A. Zakai**[1,2], **Steven D. Lidofsky**[1], **Frederick W. Unverzagt**[5], **Suzanne E. Judd**[6]

**1** Department of Medicine, Larner College of Medicine at the University of Vermont, Burlington, VT, United States of America, **2** Department of Pathology & Laboratory Medicine, Larner College of Medicine at the University of Vermont, Burlington, VT, United States of America, **3** Department of Mathematics and Statistics, University of Vermont, Burlington, VT, United States of America, **4** Division of Gerontology, Department of Medicine, Geriatrics and Palliative Care, University of Alabama at Birmingham, Birmingham, AL, United States of America, **5** Department of Psychiatry, Indiana University School of Medicine, Indianapolis, IN, United States of America, **6** Department of Biostatistics, University of Alabama at Birmingham, Birmingham, AL, United States of America

* Mary.Cushman@uvm.edu

**Data Availability Statement:** The data underlying the findings include potentially identifying participant information, and cannot be made publicly available due to ethical/legal restrictions.

## Abstract

### Background & aims

Nonalcoholic fatty liver disease (NAFLD) is prevalent and may affect cognitive function. We studied associations of NAFLD with risk of cognitive impairment. Secondarily we evaluated liver biomarkers (alanine aminotransferase (ALT), aspartate aminotransferase (AST), their ratio, and gamma-glutamyl transpeptidase).

### Methods

In a prospective cohort study, the REasons for Geographic and Racial Differences in Stroke, among 30,239 black and white adults aged ≥45,495 cases of incident cognitive impairment were identified over 3.4 years follow up. Cognitive impairment was identified as new impairment in two of three cognitive tests administered every two years during follow up; word list learning and recall, and verbal fluency. 587 controls were selected from an age, race, sex-stratified sample of the cohort. The fatty liver index was used to define baseline NAFLD. Liver biomarkers were measured using baseline blood samples.

### Results

NAFLD at baseline was associated with a 2.01-fold increased risk of incident cognitive impairment in a minimally adjusted model (95% CI 1.42, 2.85). The association was largest in those aged 45–65 (p interaction by age = 0.03), with the risk 2.95-fold increased (95% CI 1.05, 8.34) adjusting for cardiovascular, stroke and metabolic risk factors. Liver biomarkers were not associated with cognitive impairment, except AST/ALT >2, with an adjusted OR 1.86 (95% CI 0.81, 4.25) that did not differ by age.

However, data including statistical code from this paper are available to researchers who meet the criteria for access to confidential data. Data can be obtained upon request through the University of Alabama at Birmingham at regardsadmin@uab.edu.

**Funding:** This research project is supported by cooperative agreement U01 NS041588 co-funded by the National Institute of Neurological Disorders and Stroke (NINDS) and the National Institute on Aging (NIA), National Institutes of Health (NIH), Department of Health and Human Service. Authors receiving funding from this grant: MC, VW, NAZ, FWU, SEJ. Additional funding from National Heart Lung and Blood Institute of NIH to KSA: T32HL007594. Representatives of the NINDS were involved in the study design but not in the collection, management, analysis or interpretation of the data, decision to publish or preparation of the manuscript.

**Competing interests:** The authors have declared that no competing interests exist.

## Conclusions

A laboratory-based estimate of NAFLD was associated with development of cognitive impairment, particularly in mid-life, with a tripling in risk. Given its high prevalence, NAFLD may be a major reversible determinant of cognitive health.

## Introduction

Cognitive impairment is increasing with the aging of the global population, and risk factor levels at younger age contribute to this burden [1]. The prevalence of liver disease and cirrhosis is also increasing [2] and a recognized complication of the latter is hepatic encephalopathy, a condition manifested by alterations in the level of consciousness attributable to increased levels of potentially neurotoxic compounds usually filtered by the liver. Although most individuals with cirrhosis do not have overt hepatic encephalopathy, a form of subclinical neurocognitive impairment, minimal hepatic encephalopathy, is relatively common in this population [3, 4]. Whether chronic liver disease without cirrhosis affects cognitive function remains uncertain, and the pathways of an association are unclear. Simply considered, toxin accumulation might cause neuronal damage, as might lowered production of protective substances [5].

The most common form of chronic liver disease in industrialized nations is nonalcoholic fatty liver disease (NAFLD), a metabolic disorder characterized by hepatocellular fat overload and injury in the absence of significant alcohol use. NAFLD is present in 30% of American adults, and increasing with the obesity epidemic[2]. Little is known about its impact on cognitive function [6–10].

As a manifestation of the metabolic syndrome [11], NAFLD frequently occurs when there are multiple cardiovascular risk factors, including hypertension, diabetes mellitus, obesity and dyslipidemia [12]. Both metabolic syndrome and diabetes are themselves associated with cognitive decline and dementia [13]. In much older adults this relationship is reversed, perhaps related to weight loss marking other conditions associated with cognition [14–16]. Two epidemiological studies suggested that the association of NAFLD with cognitive function or relevant brain imaging parameters was greater in younger than older people [17, 18]. Consequently, we hypothesized that NAFLD would be associated with cognitive impairment and the association would be greater among younger people.

To test this, we investigated the relationship between NAFLD and incident cognitive impairment in a national US cohort, the REasons for Geographic And Racial Differences in Stroke (REGARDS). The fatty liver index (FLI), a surrogate marker of NAFLD with reasonable validity [19–21], was used to estimate NAFLD. Secondarily, we studied associations of individual liver disease biomarkers with cognitive impairment.

## Materials and methods

### Cohort

The REGARDS study is a National Institute of Neurologic Disorders and Stroke, and National Institute on Aging funded study investigating reasons for racial and regional differences in stroke mortality and cognitive impairment. Details on the study design were previously published [22]. There are 30,239 participants in REGARDS, 45 years of age and older at recruitment (2003–07), located throughout the contiguous United States, with 56% living in the stroke belt of the southeastern U.S. (North Carolina, South Carolina, Georgia, Alabama,

Mississippi, Tennessee, Arkansas, and Louisiana). After telephone-based enrollment and an in-home visit for data collection, participants were followed by telephone every 6 months. Participants self-identified as Black (41%) or White (59%) and 55% were women.

## Standard protocol approvals and patient consents

All study procedures, including the consent process, were reviewed and approved by the institutional review boards of the collaborating institutions (University of Alabama at Birmingham Institutional Review Board, University of Vermont Committee on Human Research, University of Cincinnati Institutional Review Board, Wake Forest University Institutional Review Board). Telephone interviewers were trained to identify participants who answered questions in a way that indicated lack of comprehension. Potential participants who were able to respond to telephone questions provided verbal consent, which was followed up with written consent at an in home visit.

## Cognitive assessments

Cases of incident cognitive impairment were defined as previously published using four cognitive tests administered by telephone: the Six-item Screener used at baseline [23], and an animal fluency test, a word list learning test and word list recall given in staggered fashion every two years during follow-up [24]. Test scores on each of the latter three tests were considered impaired if the most recent administration at the time of case identification for this study was more than 1.5 standard deviations below age- race- sex- and education-adjusted mean scores [25]. This definition allowed for correction for demographic factors on test performance. Incident cognitive impairment was defined as impaired scores on at least two of the three follow-up tests at the most recent administration, among participants with a normal Six-item Screener at baseline.

## Study design

As previously reported [25] we used a nested case-control study design after 3.4 years of follow up to study the relationship between biomarkers and risk of cognitive impairment. We excluded participants with baseline self-reported stroke, baseline cognitive impairment on the Six-item Screener, insufficient cognitive testing, or anomalous data. Among the remaining 17,630 participants, we identified 495 cases of incident cognitive impairment. From an 1100-person random sample of the cohort selected for a case-cohort study of stroke outcomes [26], we selected 587 unmatched controls that met the eligibility criteria applied to cases. The 1100-person random sample was selected to provide sufficient representation of each race group, both genders, and age groups 45 to 54 (20%), 55 to 64 (20%), 65 to 74 (25%), 75 to 84 (25%), and ≥85 (10%). The flow chart describing this nested case-control study sample selection was previously published [25].

## Laboratory

Baseline fasting blood samples were obtained at the in-home visit, processed, and shipped on ice overnight to the University of Vermont, where they were centrifuged and stored at -80°C [27]. Lipid profile and glucose were measured as previously reported [28]. In the case-control sample, alanine aminotransferase (ALT), aspartate aminotransferase (AST) and gamma-glutamyl transpeptidase (GGT) were measured in serum using the Roche Elecsys 2010 analyzer (Roche Diagnostics Indianapolis, IN), with analytical inter-assay CV ranges 1.1%-5.4% for ALT, 1.8%-6.8% for AST, and 0.7%-3.0% for GGT.

## Fatty liver index

The FLI is a surrogate marker for NAFLD, developed by Bedogni *et al.* for use in epidemiologic research where imaging studies or the gold standard liver biopsy, are not feasible [19]. In identifying NAFLD the FLI compares well to the presence of increased hepatic steatosis by ultrasound [20, 21, 29] and proton magnetic resonance spectroscopy [30], but it does not quantitate the extent of steatosis. The FLI is calculated using the formula:

$$\frac{e^{0.953*\log(\text{triglycerides})+0.139*\text{BMI}+0.718*\log(\text{GGT})+0.053*\text{waist circumference}-15.745}}{1+e^{0.953*\log(\text{triglycerides})+0.139*\text{BMI}+0.718*\log(\text{GGT})+0.053*\text{waist circumference}-15.745}} \times 100$$

An FLI >60 suggests NAFLD (probability to have NAFLD is 78%), and FLI <20 has a high sensitivity to rule out NAFLD (probability to not have NAFLD 91%) [19, 31]. Individuals with FLI >60 in the absence of heavy alcohol intake, as defined below, were considered to have NAFLD.

## Covariates

All covariates were based on data collected at baseline. Race, alcohol drinks per week, income, physical activity level and prebaseline stroke were established by participant self-report, and diabetes, hypertension, and dyslipidemia were defined using study measurements as previously described [32]. Left ventricular hypertrophy (LVH) was established by electrocardiogram [33] and atrial fibrillation was determined by self-report or presence on the baseline electrocardiogram. Diabetes was defined as fasting glucose ≥ 126 mg dL−1, nonfasting glucose ≥ 200 mg dL−1, or self-reported use of medications for diabetes. Dyslipidemia was defined based on definitions at REGARDS baseline as total cholesterol ≥240 mg/dL low-density lipoprotein cholesterol ≥160 mg/dL, high-density lipoprotein cholesterol ≤40 mg/dL, or self-reported current use of lipid lowering therapy. Body-mass index (BMI) was calculated using baseline measured height and weight, with weight measured using a standard 136 kilogram calibrated scale and height measured with a 2.5 meter metal tape measure and square, both measured without shoes. Alcohol use was classified as none, moderate (≤14 drinks/week for men, ≤7 drinks/week for women, or heavy (>14 drinks/week for men or >7 drinks/week for women). Education was categorized as <high school, some college, and college graduate or higher. Yearly income was categorized as <$20,000, $20,000–34,999, $35,000–74,999, >$75,000 or refused to respond. Physical activity was categorized as any weekly exercise or none by response to the question, "How often each week to you exercise enough to work up a sweat" [34]. Baseline cardiovascular disease was defined as electrocardiogram evidence of myocardial infarction or self-reported myocardial infarction, coronary artery bypass, angioplasty, stent, or peripheral artery disease.

## Statistical analyses

Statistical analyses were performed with SAS 9.3 (Cary, NC). Baseline participant characteristics by NAFLD status and case control status were tabulated with weighting based on age group, sex, race to account for the stratified control sample selection so that the characteristics would reflect those in the larger REGARDS cohort.

To estimate relative risk, odds ratios (OR) of incident cognitive impairment for NAFLD and liver biomarkers were calculated using weighted logistic regression models. For analysis of NAFLD those with heavy alcohol use were excluded. Multilevel categorical variables were assessed as indicator variables. Model 1 adjusted for baseline age, sex, region of residence and race. Model 2 additionally adjusted for baseline education and income. Model 3 additionally

adjusted for baseline systolic blood pressure, LVH, smoking, cardiovascular disease, atrial fibrillation, diabetes and hypertension medication use. Model 4 added baseline alcohol use, BMI, and physical activity to Model 3. We analyzed the liver markers AST, ALT, GGT in sex-specific quartiles based on their distribution in controls and per standard deviation (SD) increments. We assessed AST/ALT ratio, because a ratio greater than 2 is associated with long-term complications from chronic liver disease [35]. Interactions with age, race and sex were tested using cross-product terms with p <0.10 for the interaction term considered significant for NAFLD status and P <0.05 for individual liver markers (a more stringent threshold given multiple tests). The study had 90% power to detect on OR 1.45 for cognitive impairment with NAFLD with alpha 0.05.

## Results

### Participant characteristics

Overall there were 495 cases of incident cognitive impairment and 587 controls (representing 17,135 REGARDS participants eligible to become a case). Fig 1 shows the inclusion of cases and controls in the analysis of FLI and risk of cognitive impairment, and the proportion of each group classified with NAFLD. Few participants were heavy alcohol users; missing status for FLI related primary to missing stored blood samples.

Table 1 shows that those with NAFLD were more likely men, Black persons, were less well educated, had adverse cardiovascular risk profiles and higher ALT and GGT. They did not differ by age, alcohol use, income or prevalent cardiovascular disease. Table 2 presents the distribution of cardiovascular risk factors, NAFLD, and liver biomarkers in participants with incident cognitive impairment and controls. Cases were more likely to live in the stroke belt, have lower income, and had higher BMI, GGT, and blood pressure than controls and were more likely than controls to be smokers, and have diabetes, cardiovascular disease, and NAFLD. They were less likely to have moderate alcohol consumption.

### Associations of NAFLD with cognitive impairment

NAFLD was significantly associated with incident cognitive impairment in the minimally-adjusted Model 1 (OR: 2.01; 95% CI: 1.42, 2.85). There were no significant differences in this

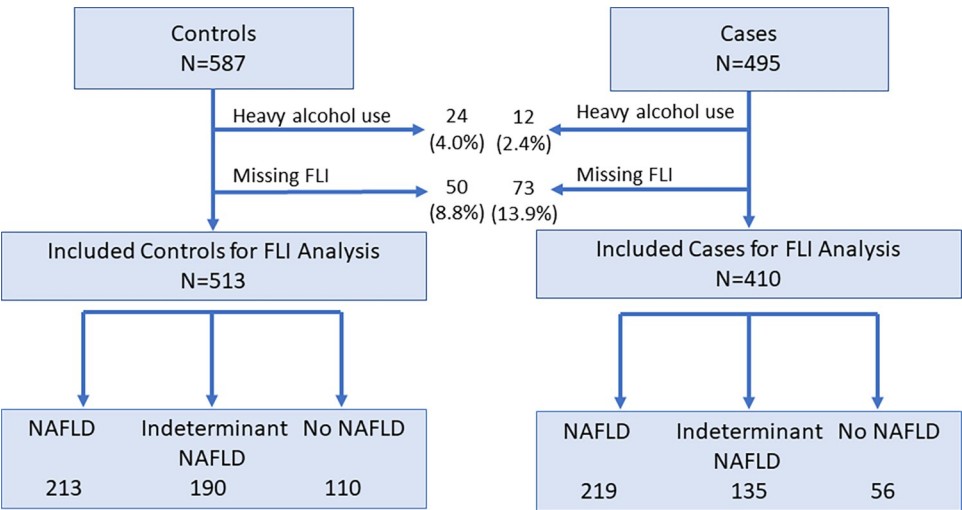

**Fig 1. Inclusion of cases and controls in the analysis of the association of FLI with risk of incident cognitive impairment.**

**Table 1. Baseline characteristics\* by NAFLD status in the control group.**

| Characteristic, mean or percent | NAFLD | No NAFLD |
|---|---|---|
| | (FLI >60, n = 213; weighted n = 6476) | (FLI <20, n = 110; weighted n = 3031) |
| Women, % | 48 | 74 |
| Black, % | 39 | 29 |
| Age, mean | 63.7 | 64.2 |
| Region, % | | |
| Stroke belt | 51 | 53 |
| Nonbelt | 49 | 47 |
| Education <high school, % | 10 | 2 |
| Income <$20,000 | 12 | 11 |
| Hypertension, % | 59 | 35 |
| Left ventricular hypertrophy, % | 9 | 1 |
| Systolic Blood Pressure, mean | 130 | 120 |
| Body-mass Index, mean kg/m$^2$ | 33.4 | 23.1 |
| Current smoking, % | 14 | 11 |
| Exercise, % none | 34 | 23 |
| Dyslipidemia, % | 69 | 34 |
| Diabetes, % | 30 | 5 |
| Prevalent cardiovascular disease, % | 15 | 11 |
| Alcohol use, % moderate | 40 | 40 |
| AST, mean U/L | 22 | 20 |
| ALT, mean U/L | 21 | 14 |
| GGT, mean U/L | 40 | 18 |

\* Definitions of variables can be found in the methods section.

Abbreviations: NAFLD, Nonalcoholic fatty liver disease; FLI, fatty liver index; AST, aspartate aminotransferase; ALT, alanine aminotransferase; GGT, gamma-glutamyl transpeptidase

relationship by sex (p interaction 0.45) or race (p interaction 0.44), however, as shown in Table 3, there was a nearly 4-fold increased risk in those under age 65 and no association above age 65 (p for age interaction 0.03). Results were similar across the four adjusted models, with modest attenuation in the fully adjusted Model 4 (OR 2.95; 95% CI 1.05, 8.34).

## Associations of liver biomarkers with cognitive impairment

Fig 2 demonstrates that, after adjustment for risk factors (Model 4), none of the liver biomarkers were significantly associated with cognitive impairment, but those in the 2$^{nd}$ and 3$^{rd}$ quartiles of all three markers appeared to have lower risk. This U-shaped association in the fully adjusted models was not statistically significant except for GGT (p value for an added quadratic term for AST was 0.72, for ALT 0.12 and for GGT 0.03). None of the biomarkers differed in their association with cognitive impairment by race, sex, or age (all p interactions >0.05), except GGT, which interacted with age (p = 0.02). Unlike for NAFLD, stratified analyses of GGT and cognitive impairment by age groups did not reveal a pattern explaining this interaction.

Finally, as shown in Table 4, AST/ALT ratio >2 was associated with a 2-fold higher risk of incident cognitive impairment in the minimally adjusted model (OR 2.05; 95% CI 1.16, 3.60), and this relationship was mildly attenuated adjusting for risk factors (OR 1.86; 95% CI 0.81,

**Table 2. Levels of risk factors, NAFLD, and liver biomarkers in incident cognitive impairment cases and controls.**

| Characteristic, mean (SD) or percent | Cognitive Impairment Cases | Controls |
|---|---|---|
| | (n = 495) | (weighted n = 17,135) |
| Women, % | 59 | 57 |
| Black, % | 33 | 36 |
| Age, mean | 64.6 (10.2) | 64.1 (8.7) |
| Region, % | | |
| Stroke Belt | 66 | 52 |
| Non Stroke Belt | 35 | 48 |
| Education <high school, % | 8 | 7 |
| Income <$20,000 | 25 | 12 |
| Hypertension, % | 55 | 51 |
| Left ventricular hypertrophy, % | 12 | 7 |
| Systolic Blood Pressure, mean | 128 | 126 |
| Body-mass Index, mean kg/m$^2$ | 30.0 (6.1) | 29.2 (5.6) |
| Current smoking, % | 17 | 12 |
| Exercise, % none | 38 | 32 |
| Dyslipidemia, % | 58 | 56 |
| Diabetes, % | 28 | 18 |
| Prevalent cardiovascular disease, % | 21 | 14 |
| Alcohol use, % | | |
| Heavy | 2 | 5 |
| Moderate | 28 | 34 |
| NAFLD Status† | | |
| NAFLD, % FLI >60 | 54 | 43 |
| NO NAFLD, % FLI <20 | 14 | 21 |
| AST, mean U/L† | 21 | 21 |
| ALT, mean U/L† | 18 | 18 |
| GGT, mean U/L† | 38 | 30 |

†FLI was missing in 128 participants (76 cases, 52 controls) AST in 123 (78 cases, 45 controls), ALT in 117 (75 cases, 42 controls), and GGT in 111 (72 cases, 39 controls). Primary reasons for missing data were absence of blood samples, which was missing at random.

Abbreviations: NAFLD, Nonalcoholic fatty liver disease; FLI, fatty liver index; AST, aspartate aminotransferase; ALT, alanine aminotransferase; GGT, gamma-glutamyl transpeptidase

4.25 in Model 4). There were no differences in this association by age, sex, or race (all p >0.50).

## Discussion

Here we demonstrate that NAFLD is a risk factor for incident cognitive impairment in a US population sample. As classified by the FLI, NAFLD was associated with a tripling in the risk of cognitive impairment among participants aged 45–65 with no association in older individuals up to age 100 at baseline. None of the individual liver biomarkers studied were associated with cognitive impairment, although AST/ALT ratio had a suggestive association that did not differ by age, race or sex. Findings add to a growing body of evidence that risk factors present at younger ages are important determinants of cognitive function [1].

Most research on the cognitive impacts of liver disease focused on effects of relatively severe chronic or acute liver dysfunction [4]. In healthier populations, some but not all cross-

**Table 3. Odds ratio (95% confidence interval) of cognitive impairment by NAFLD.**

| Model† | Age-Stratified OR of Cognitive Impairment by FLI >60 vs <20‡ | | |
| --- | --- | --- | --- |
| | < 65 years | 65-<75 years | 75+ years |
| | n = 213 (7807)§ | n = 130 (5113)§ | n = 67 (1925)§ |
| 1 | 3.84 (2.12, 6.96) | 1.33 (0.61, 2.90) | 0.76 (0.30, 1.91) |
| 2 | 4.63 (2.38, 8.99) | 1.24 (0.49, 3.10) | 0.87 (0.37, 2.43) |
| 3 | 3.60 (1.54, 8.42) | 1.16 (0.36, 3.69) | 0.73 (0.18, 3.01) |
| 4 | 2.95 (1.05, 8.34) | 1.20 (0.23, 6.32) | 0.76 (0.09, 6.72) |

Abbreviations: NAFLD, Nonalcoholic fatty liver disease; FLI, fatty liver index

† Model 1: adjusted for age, race, region, and sex

Model 2: additionally adjusted for education and income

Model 3: additionally adjusted for systolic blood pressure, left ventricular hypertrophy, smoking, prebaseline cardiovascular disease, atrial fibrillation, diabetes, and hypertension medication use

Model 4: additionally adjusted for alcohol use, body-mass index, and physical activity

‡ P interaction for NAFLD and age in Model 1 = 0.03

§ weighted n of cases (noncases)

sectional studies indicated that individuals with NAFLD may have more cognitive impairment or dementia than controls [8, 9, 36–39], and in a recent systematic review and meta-analysis of this topic, only one prospective study was available, and this was a small clinic population of 331 people with NAFLD and alcoholic liver disease who were followed over 3 years for functional difficulty and did not have follow up cognitive testing or a control group, so did not address the question of the relationship of NAFLD to incident cognitive impairment [10]. A subsequently published study from China involving 1,651 community-based participants followed for 4 years demonstrated a similar association of NAFLD with incident cognitive impairment (based on the mini-mental state exam only) as we observed here, with this association limited to those <65 years at baseline and no association in older people [18]. The Framingham Heart Study investigators provided relevant clinicopathological information, demonstrating a cross-sectional association of NAFLD with lower cerebral brain volume (but not hippocampal or white matter hyperintensity volumes) after adjustment for a variety of relevant confounders in cognitively normal participants [17]. Notably, similar to here, associations were larger in participants younger than 60 years of age. Considering a subset of 1287 well characterized Framingham participants (378 with NAFLD by imaging) there were no cross-sectional associations of NAFLD with test scores evaluating memory, abstract reasoning, visual perception, attention and executive function [39]. As NAFLD frequently occurs with obesity and diabetes, it can be difficult to separate the effects of liver steatosis from that of other components of the metabolic syndrome, which are associated with declines in cognition. In the current analysis adjustment for diabetes, body-mass index and a variety of other factors, did not confound the association of NAFLD with cognitive impairment. While residual confounding could be present, findings support the need to study reasons for this independent association.

Our finding that NAFLD was a stronger risk factor for development of cognitive impairment in participants <65 is consistent with the two studies mentioned above [17, 18], and with a number of studies showing that while obesity and the metabolic syndrome are risk factors for dementia in younger individuals [40–42], this situation reverses later in life. Similar to here, one large study also reported a positive association of NAFLD by the FLI with cardiovascular disease in younger people, and an inverse association in older people [43]. Prospective studies in individuals age 65 and older in the U.S. [44, 45] and Australia [46], showed an

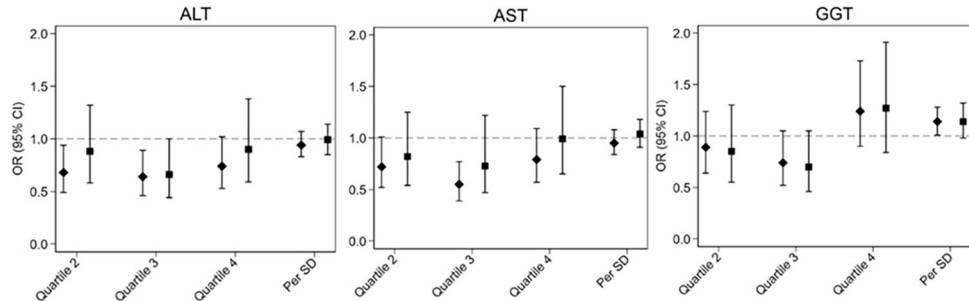

**Fig 2. Odds ratios of cognitive impairment by AST, ALT, and GGT.** ORs were calculated based on quartiles, comparing each higher quartile to the first quartile, and per SD increment. Quartiles were defined based on the distribution in controls. Diamond-shaped symbols denote Model 1 and squares denote Model 4 odds ratios. See Tables 3, 4 for model covariates.

Quartile cut-off values.

|  | 25% | 50% | 75% |
|---|---|---|---|
| AST |  |  |  |
| Men | 17 | 20 | 24 |
| Women | 15 | 18 | 22 |
| ALT |  |  |  |
| Men | 13 | 17 | 23 |
| Women | 10 | 14 | 17 |
| GGT |  |  |  |
| Men | 18 | 25 | 37 |
| Women | 13 | 18 | 27 |

Abbreviations: AST, aspartate aminotransferase; ALT, alanine aminotransferase; GGT, gamma-glutamyl transpeptidase.

inverse relationship between BMI and cognitive function, and a study of participants age 85 and older in the Netherlands [15] demonstrated a reduced risk of cognitive decline in those with the metabolic syndrome. Our results indicate that NAFLD, a manifestation of the metabolic syndrome, follows a similar pattern in its associations with cognitive impairment and with cardiovascular disease as reported by others [43]. These patterns from multiple studies may be due to competing risks for other outcomes, especially mortality, at older age. Also, in

**Table 4. Association of AST/ALT >2 with incident cognitive impairment.**

| Model† | OR (95% CI) |
|---|---|
| 1 | 2.05 (1.16, 3.60) |
| 2 | 1.72 (0.92, 3.21) |
| 3 | 1.88 (0.85, 4.13) |
| 4 | 1.86 (0.81, 4.25) |

Abbreviations: AST, aspartate aminotransferase; ALT, alanine aminotransferase

† Model 1: adjusted for age, race, region, and sex

Model 2: additionally adjusted for education and income

Model 3: additionally adjusted for systolic blood pressure, left ventricular hypertrophy, smoking, prebaseline cardiovascular disease, atrial fibrillation, diabetes, and hypertension medication use

Model 4: additionally adjusted for alcohol use, body-mass index and physical activity

younger people there may be fewer adverse mechanisms affecting cognition, thus an association of NAFLD is detectable. At older age, multiple independent adverse pathways lead to cognitive impairment [47], likely muting any association of NAFLD with cognition.

Little is known about the relationship between the liver biomarkers AST, ALT, and GGT and cognition. Some evidence is emerging for GGT, a liver marker with pro-oxidant and inflammatory properties. In a cross-sectional study, GGT and AST were correlated with visual attention and verbal memory in veterans with alcohol dependence and post-traumatic stress disorder [48]. Higher GGT was associated with dementia in middle- and older-age Finnish men [49], and with cognitive decline and vascular dementia after age 80 [50], but genetic variation of GGT was not associated with risk of Alzheimer's disease in the large International Genomics of Alzheimer's Project, suggesting no causal relationship [51].

Despite no relationship of the individual biomarkers with incidence of cognitive impairment in this general population study, we observed a suggestive association of AST/ALT ratio. A recent report from the Alzheimer's Disease Neuroimaging Initiative showed cross-sectional associations of elevated ASL/ALT ratio with a diagnosis of Alzheimer's disease, and including patients with or without cognitive disorders, with poorer cognitive function test scores and positron emission tomography and cerebrospinal fluid biomarkers of amyloid, tau and neurodegeneration [52]. The only covariates considered in this study were age, sex, body mass index (BMI), and APOE ε4 status, so it is not clear that associations were independent of socioeconomic factors, alcohol use or cerebrovascular mechanisms. In the current study, socioeconomic factors had the largest attenuating effect on the association of AST/ALT with cognitive impairment (OR 2.05 to 1.72 after accounting for this). We did not account for APOE ε4 as it was not associated with our outcome of cognitive impairment. Nonetheless, together with findings from this prospective study, a causal relationship of liver disease and risk of cognitive impairment may be hypothesized. Increased AST/ALT ratio can be associated with alcohol damage and it can be observed with cirrhosis as well. However, cognitive impairment was not was not higher with alcohol use in REGARDS [53], and those with heavy alcohol use were excluded in the current study. It is thus possible that our findings reflect a contribution to cognitive impairment from unrecognized NAFLD related cirrhosis. Further investigation to test this would be appealing.

The main strength of this study is the use of a large biracial population-based prospective study, which provided nearly 500 well characterized incident cognitive impairment cases. Our definition of cognitive impairment accounted for the effects of age, race, sex, and education, so accounted for these factors. It also included testing in multiple cognitive domains, expanding on the little prior prospective data [18]. The main limitation of this work is the use of the FLI as a surrogate marker for NAFLD. While the FLI, unlike gold standards, cannot quantitatively assess liver fat, it compares well with ultrasound and proton magnetic resonance spectroscopy determination of steatosis [21, 30], and has been related to risk of other outcomes like cardiovascular disease, hypertension and diabetes [43, 54, 55]. Consistency of our findings with others also supports validity of the FLI. Whether participants had cirrhosis or liver fibrosis is unknown. FLI presence may also relate closely to metabolic syndrome, and as such it would be difficult to separate the impacts of NAFLD and metabolic syndrome on risk of cognitive impairment. While we adjusted for factors involved in metabolic syndrome, we cannot say definitively that NAFLD is causal for the observed associations. While we excluded people with heavy alcohol use from analyses of NAFLD as an exposure, it remains possible some participants classified with NAFLD had alcohol-related liver disease. Generalizability of the study sample was limited to Black and White persons in the US, so results require replication in other contexts. While the classification used for cognitive impairment has clinical relevance and is likely to identify those with significant impairment, telephone-based tests may lead to

misclassification, and we do not know the sensitivity and specificity of our definition for dementia (especially in younger people). Dementia outcomes are not yet available in REGARDS. It is possible that associations of NAFLD with cognitive impairment were not apparent in older participants due to differential loss to follow up based on cognitive outcome, or unmeasured confounders impacting on the relationship between NAFLD and cognition in older people. It is also possible that the measures we used are more sensitive to detecting abnormal cognitive dysfunction related to NAFLD in younger than older people because there are less pathways to cognitive dysfunction in younger people.

In summary, this research demonstrated a tripling of the risk of incident cognitive impairment in middle-to-early-older age Black and White US residents with NAFLD. There were no associations for ALT, AST or GGT individually, and additional research is needed concerning the significance of the AST/ALT ratio findings. More study is also needed to determine the mechanisms behind our observations. If these findings are further confirmed, including in studies with better classification of NAFLD than FLI, given its high prevalence in western countries, NAFLD may be a major reversible determinant of impaired cognitive health.

## Acknowledgments

The authors thank the other investigators, the staff, and the participants of the REGARDS study for their valuable contributions. A full list of participating REGARDS investigators and institutions can be found at: https://www.uab.edu/soph/regardsstudy/.

## Author Contributions

**Conceptualization:** Mary Cushman, Virginia Wadley, Neil A. Zakai, Frederick W. Unverzagt, Suzanne E. Judd.

**Data curation:** Virginia Wadley, Frederick W. Unverzagt, Suzanne E. Judd.

**Formal analysis:** Peter W. Callas, Kristine S. Alexander.

**Funding acquisition:** Mary Cushman.

**Investigation:** Mary Cushman, Virginia Wadley, Frederick W. Unverzagt, Suzanne E. Judd.

**Methodology:** Peter W. Callas, Neil A. Zakai, Steven D. Lidofsky, Suzanne E. Judd.

**Project administration:** Suzanne E. Judd.

**Writing – original draft:** Mary Cushman.

**Writing – review & editing:** Peter W. Callas, Kristine S. Alexander, Virginia Wadley, Neil A. Zakai, Steven D. Lidofsky, Frederick W. Unverzagt, Suzanne E. Judd.

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
