## [Decision Letter · Decision Letter 0]

31 May 2022

PONE-D-22-08991Nonalcoholic Fatty Liver Disease and Cognitive Impairment: a Prospective Cohort StudyPLOS ONE

Dear Dr. Cushman,

Thank you for submitting your manuscript to PLOS ONE. After careful consideration, we feel that it has merit but does not fully meet PLOS ONE’s publication criteria as it currently stands. Therefore, we invite you to submit a revised version of the manuscript that addresses the points raised during the review process.

 The revised manuscript should provide clarity on additional points as summarized by Reviewer 1 with additional stylistic and reference considerations from Reviewer 2.

We look forward to receiving your revised manuscript.

Kind regards,

Nicholette D. Palmer, Ph.D.

Academic Editor

PLOS ONE

Journal Requirements:

Reviewers' comments:

Reviewer's Responses to Questions

**Comments to the Author**

1. Is the manuscript technically sound, and do the data support the conclusions?

Reviewer #1: Yes

Reviewer #2: Yes

2. Has the statistical analysis been performed appropriately and rigorously? 

Reviewer #1: Yes

Reviewer #2: I Don't Know

3. Have the authors made all data underlying the findings in their manuscript fully available?

Reviewer #1: Yes

Reviewer #2: Yes

4. Is the manuscript presented in an intelligible fashion and written in standard English?

Reviewer #1: Yes

Reviewer #2: Yes

5. Review Comments to the Author

Reviewer #1: Overall

This manuscript overall addresses a novel and interesting research question. Overall, the analysis has been well thought out and executed. I think with some changes to clarify details throughout (detailed below) and some clarity around the primary and secondary outcomes, and toning back the conclusions, this would be a publishable piece of work.

Abstract

Background/aims- Would clearly indicate primary aim and secondary aims (as all outcomes are listed together currently.

Methods- Were the controls derived from the same cohort? This isn’t clear

Results- NAFLD ‘at baseline’ was associated with a 2.01 fold…. I think you should be transparent as you don’t have follow up information regarding est NAFLD status

Conclusion- given you only have an est of NAFLD (no liver outcomes just estimates) and this is not a generalisable cohort I think your conclusion is an overstatement of your findings. Can you re word it to reflect these points?

Intro

Line 68/69- toxin production in chronic liver disease? Is this a likely stipulation?

Line 80- ‘the very old’ do you mean older adults? And this sentence needs to be reworded or better explained as I am not sure what you mean

Line 90- I don’t think FLI is a valid marker or NAFLD, rather a surrogate marker that has been shown to have reasonable validity in epidemiological studies. I would suggest you say it was used to ‘estimate’ NAFLD.

Line 87-90- your primary aim/ research question is not clear, can you phrase this as a primary outcome and X Y X (secondary outcomes) will also be assessed?

Methods

Line 155 – can you explain why you used a cut off below 20 not 30 for ruling out FLI?

Line 165-166- can you describe how alcohol intake was measured?

159- there are no descriptions of the methods or tools used to measure/ height, weight, physical activity etc

As per the STROBE guidelines, it would be helpful to include a flow chart describing the participant inclusion

Results

Definitions for exercise, dyslipidaemia, diabetes etc have not been made clear for table 1

I would consider combining Tables 1 and 2 as the variables are the same

Line 222- whats the definition of older age?

Discussion

Line 268- general ‘US’ population? The current wording is misleading

Line 279- and what did this study show?

Line 295- Given you adjusted for T2D, BMI and other risk factors and that did not explain your results, what does that suggest? This point is not complete

Line 310-311- this is a hypothesis? It is written as a fact and there is no reference?

319-320- your concluding sentence does not support the evidence you have presented in this paragraph

325- what are ‘patients who were normal’ what is normal?

Line 360- NAFLD – typo, and ‘strong associations’ is an overstatement as you need to state this is in a US population. I suggest you add the generalisability of your sample as a limitation further up and modify your conclusion to reflect this

Line 365- I think you can say liver specific outcomes rather than better classification which is unspecific

Line 365- NAFLD typo

Reviewer #2: Dear Authors, I was pleased to read your manuscript. The authors wanted to see if there was a link between NAFLD (as measured by the computed FLI) and cognitive impairment as measured by a verbal fluency test, a word list learning test, and word list memory during phonecall follow-up visits in the REGARDS cohort. The findings were consistent with NAFLD playing a role in the development of cognitive impairment in participants aged 45 to 65.

The study's value comes in its huge sample size, although the NAFLD and cognitive assessments are inaccurate because to REGARDS cohort restrictions. The hypothesis appears to be supported by statistics and findings. In fact, the FLI is a method to calculate the risk of having hepatic steatosis, but it does not discriminate whether the cause is due to alcohol consumption or not. Even if you have corrected for this covariate, you should state this limitation of the FLI in your discussions. Likewise, you should state that the incidence of cognitive impairment is calculated on the basis of telephone tests and is not methodologically strong.

Your discussion would probably benefit from a recently published article on the subject. In the observed cohort, the clinical diagnosis of dementia was used and these results could support yours, albeit in a different population doi: 10.3389/fnagi.2021.748888.

Figure 1 is actually a table, which is also very impractical for the average reader, in my opinion. I suggest you revise and edit.

6. PLOS authors have the option to publish the peer review history of their article (what does this mean?). If published, this will include your full peer review and any attached files.

Reviewer #1: No

Reviewer #2: No

---

## [Author Response · Author response to Decision Letter 0]

9 Nov 2022

Response to Reviewers, PONE-D-22-08991

Reviewer #1: Overall

This manuscript overall addresses a novel and interesting research question. Overall, the analysis has been well thought out and executed. I think with some changes to clarify details throughout (detailed below) and some clarity around the primary and secondary outcomes, and toning back the conclusions, this would be a publishable piece of work.

We are grateful for the effort and detailed comments from the reviewer – thank you.

Abstract

Background/aims- Would clearly indicate primary aim and secondary aims (as all outcomes are listed together currently.

We modified the abstract for clarity: 

Abstract: “We studied associations of NAFLD with risk of cognitive impairment. Secondarily we evaluated liver biomarkers (alanine aminotransferase (ALT), aspartate aminotransferase (AST), their ratio, and gamma-glutamyl transpeptidase).” 

Methods- Were the controls derived from the same cohort? This isn’t clear

We specified that controls were from the cohort: “587 controls were selected from an age, race and sex-stratified cohort sample of the cohort.”

Results- NAFLD ‘at baseline’ was associated with a 2.01 fold…. I think you should be transparent as you don’t have follow up information regarding est NAFLD status

We did not study risk of NAFLD over time, nor did we suggest that the best we can tell; we defined it in the methods section as being measured at baseline. The purpose of this study was to prospectively evaluate baseline NAFLD with future cognitive impairment, not change in NAFLD. The prospective design is a major strength of the study and reduces the possibility of reverse causation. Lack of data on incidence of NAFLD is a limitation of the study that is common to most observational cohort studies, and is beyond the scope of this analysis as we don’t have information to study this. If we could measure NAFLD during follow up and evaluate it as a time-varying risk factor, we would likely observe a larger association of NAFLD with cognitive impairment due to improved classification of NAFLD status. We clarified that NAFLD was measured at baseline.

Conclusion- given you only have an est of NAFLD (no liver outcomes just estimates) and this is not a generalisable cohort I think your conclusion is an overstatement of your findings. Can you re word it to reflect these points?

-We reworded the conclusions to be more tempered as suggested, “A laboratory-based estimate of NAFLD was associated with contributes to development of cognitive impairment,…..”

-For the information of the reviewer, the study population is quite generalizable to age-similar Black and White adults in the US. In fact, they are more similar to US Census-captured people than Black and White participants of similar age in NHANES, which is sampled purposefully to represent the general US population. Nevertheless, we added a generalizability limitation to the Discussion section as requested below by the reviewer (since we didn’t study all possible people).

Intro

Line 68/69- toxin production in chronic liver disease? Is this a likely stipulation?

We do not understand the question. In the parenthetical phrase, we present two thoughts about why liver disease might relate to cognitive function. We introduced the word, neuronal to hopefully make it more clear: “toxin accumulation might cause neuronal damage, as might lowered production of protective substances.” Line 69

Line 80- ‘the very old’ do you mean older adults? And this sentence needs to be reworded or better explained as I am not sure what you mean

 This was rephrased: “In much older adults this relationship is reversed..”

Line 90- I don’t think FLI is a valid marker or NAFLD, rather a surrogate marker that has been shown to have reasonable validity in epidemiological studies. I would suggest you say it was used to ‘estimate’ NAFLD.

Agreed; the wording was changed: “The fatty liver index (FLI), a surrogate marker of NAFLD with reasonable validity [18-20], was used to estimate NAFLD.”

Line 87-90- your primary aim/ research question is not clear, can you phrase this as a primary outcome and X Y X (secondary outcomes) will also be assessed?

Thank you - We modified the text for clarity: “To test this, we investigated the relationship between NAFLD and incident cognitive impairment in a national US cohort, the REasons for Geographic And Racial Differences in Stroke (REGARDS). The fatty liver index (FLI), a surrogate marker of NAFLD with reasonable validity [18-20], was used to estimate NAFLD. Secondarily, we studied associations of individual liver disease biomarkers with cognitive impairment.”

Methods

Line 155 – can you explain why you used a cut off below 20 not 30 for ruling out FLI?

We used a value of 20 as it is more restrictive to rule out NAFLD and we wanted to minimize misclassification as much as possible to define a reference group without NAFLD. According to a validation study by Bedogni and colleagues, if FLI is less than 20 the likelihood of not having NAFLD is greater than 91%. This value was used in at least one other epidemiology study that we cited (Gastaldelli, reference 31). We added reference 19 cited just above in this sentence. We added the rationale to the sentence: “An FLI >60 suggests NAFLD (probability to have NAFLD is 78%), and FLI <20 has a high sensitivity to rule out NAFLD (probability to not have NAFLD 91%.” Line 157.

Line 165-166- can you describe how alcohol intake was measured?

This was measured by self-report as stated (now line 164).

159- there are no descriptions of the methods or tools used to measure/ height, weight, physical activity etc

We had cited the source paper for this information (to minimize text length) but are now providing more information:

-Clarified which variables were self reported: “All covariates were based on data collected at baseline. Race, alcohol drinks per week, income, physical activity level and prebaseline stroke were established by participant self-report” line 164

-Height and weight were measured with standardized methods: “Body-mass index (BMI) was calculated using baseline measured height and weight, with weight measured using a standard 136 kilogram calibrated scale and height measured with 2.5 meter metal tape measure and square, both measured without shoes.” Line 173

-Physical activity was categorized as any weekly exercise or none by response to the question, “How often each week to you exercise enough to work up a sweat.” Line 180. We note that it has been remarkable in REGARDS papers that responses to this question identify people at risk of a variety of outcomes.

As per the STROBE guidelines, it would be helpful to include a flow chart describing the participant inclusion

We agree this is important. For selection of this nested case control study sample, this diagram was already published so we now refer the reader to the source, line 139: “The flow chart describing this nested case-control sample selection was previously published. [25] In addition, we added new Figure 1 to illustrate the inclusion of participants into the analysis of the association of FLI with risk of incident cognitive impairment.

New figure legend: Fig 1: Inclusion of cases and controls in the analysis of the association of FLI with risk of incident cognitive impairment.

New text, line 207: Fig 1 shows the inclusion of cases and controls in the analysis of FLI and risk of cognitive impairment, and the proportion of each group classified with NAFLD during follow up. Few participants were heavy alcohol users; missing status for FLI related primary to missing stored blood samples.

We removed previous text in this location that referred to the numbers in this figure.

Results

Definitions for exercise, dyslipidaemia, diabetes etc have not been made clear for table 1

-We had cited a source paper for these in the methods section, but now add the above definitions of covariates, and these, to the methods section starting on line 168: “Diabetes was defined as fasting glucose ≥ 126 mg dL−1, nonfasting glucose ≥ 200 mg dL−1, or self-reported use of medications for diabetes. Dyslipidemia was defined based on definitions at REGARDS baseline as total cholesterol ≥240 mg/dL low-density lipoprotein cholesterol ≥160 mg/dL, high-density lipoprotein cholesterol ≤40 mg/dL, or self-reported current use of lipid lowering therapy.” 

-A footnote was added to table 1, “* Definitions of variables can be found in the methods section.

I would consider combining Tables 1 and 2 as the variables are the same

We prefer to keep the tables separate as Table 1 only refers to the control group baseline characteristics by NAFLD status and Table 2 refers to characteristics based on presence or absence outcomes. A merged table would be a bit unwieldy and large. We only have 4 tables. If the editor prefers, we can move Table 2 to supplemental material.

Line 222- whats the definition of older age? 

This referred to people older than 65 and we thought it might be obvious from the context of the sentence. We now clarify: “…there was a nearly 4-fold increased risk in those under age 65 and no association above age 65 (p for age interaction 0.03).” Line 239.

Discussion

Line 268- general ‘US’ population? The current wording is misleading

Good point. While REGARDS is a good representation of age-similar Black and White Americans, we haven’t published this evidence; we removed the word general.

Line 279- and what did this study show?

Thank you for asking. The study actually did not assess cognitive change. The sentence was edited: “only one prospective study was available, and this was a small clinic population of 331 people with NAFLD and alcoholic liver disease who were followed over 3 years for functional difficulty and did not have follow up cognitive testing or a control group, so did not address the question of the relationship of NAFLD to incident cognitive impairment.” Line 297.

Line 295- Given you adjusted for T2D, BMI and other risk factors and that did not explain your results, what does that suggest? This point is not complete

We clarified that the purpose of adjustment was to control for confounding. We also added another sentence following, “While residual confounding could be present, findings support the need to study reasons for this independent association.” Line 315

Line 310-311- this is a hypothesis? It is written as a fact and there is no reference?

 Reference 47 was added.

319-320- your concluding sentence does not support the evidence you have presented in this paragraph

The sentence was removed.

325- what are ‘patients who were normal’ what is normal?

 This was rephrased, “including patients with or without cognitive disorders,….” Line 345

Line 360- NAFLD – typo, and ‘strong associations’ is an overstatement as you need to state this is in a US population. I suggest you add the generalisability of your sample as a limitation further up and modify your conclusion to reflect this

-The sentence was rephrased” “In summary, this research demonstrated a tripling of the risk of incident cognitive impairment in middle-to-early-older age Black and White US residents with NAFLD.” Line 383.

-We added the requested limitation on line 372: ” Generalizability of the study sample was limited to Black and White persons in the US, so results require replication in other contexts.” 

-We deleted a related sentence that was already present and was aimed at addressing this point, but didn’t use the term generalizability, “Finally, we cannot make conclusions about populations that were not studied, such as other race/ethnic groups or younger people.”

-The reviewer may also have missed that the conclusions already called for more studies including replication as a means to address generalizability and confidence in the findings.

Line 365- I think you can say liver specific outcomes rather than better classification which is unspecific

The reviewer is asking us to consider liver disease as an outcome. Our point is to consider liver disease as an exposure which is measured using better methods than we were able to apply. We tried to clarify the meaning of the sentence, “If these findings are further confirmed in studies with better classification of NAFLD than FLI….” Line 388

Line 365- NAFLD typo

 Corrected

Reviewer #2: Dear Authors, I was pleased to read your manuscript. The authors wanted to see if there was a link between NAFLD (as measured by the computed FLI) and cognitive impairment as measured by a verbal fluency test, a word list learning test, and word list memory during phonecall follow-up visits in the REGARDS cohort. The findings were consistent with NAFLD playing a role in the development of cognitive impairment in participants aged 45 to 65.

 Thank you for these comments.

The study's value comes in its huge sample size, although the NAFLD and cognitive assessments are inaccurate because to REGARDS cohort restrictions. The hypothesis appears to be supported by statistics and findings. In fact, the FLI is a method to calculate the risk of having hepatic steatosis, but it does not discriminate whether the cause is due to alcohol consumption or not. Even if you have corrected for this covariate, you should state this limitation of the FLI in your discussions. 

We recognize that the FLI does not distinguish NAFLD from alcoholic liver disease. To address this, we had excluded participants with heavy alcohol use from all analyses of NAFLD as an exposure (defined as >14 drinks/week for men or >7 drinks/week for women) to minimize any impact of alcohol-related liver disease. We added a sentence to the limitations section on line 366: “While we excluded people with heavy alcohol use from analyses of NAFLD as an exposure, it remains possible some participants classified with NAFLD had alcohol-related liver disease.” Line 370.

Likewise, you should state that the incidence of cognitive impairment is calculated on the basis of telephone tests and is not methodologically strong.

We had cited evidence that the telephone-administered tests have validity (references 23 and 24), but nevertheless added a statement about this on line 375: “telephone-based tests may lead to misclassification.” We believe the large cohort size here and prospective study design mitigates many of the limitations, including this.

Your discussion would probably benefit from a recently published article on the subject. In the observed cohort, the clinical diagnosis of dementia was used and these results could support yours, albeit in a different population doi: 10.3389/fnagi.2021.748888.

 Thank you. We added this cross-sectional study to the discussion.

Figure 1 is actually a table, which is also very impractical for the average reader, in my opinion. I suggest you revise and edit.

We think the reviewer is mistaken. Figure 1 includes 3 panels that are graphs. The reviewer must be looking at the figure legend in the text which contains tabular items describing the data (the actual figure is found elsewhere in the PlosOne system).

---

## [Decision Letter · Decision Letter 1]

19 Dec 2022

PONE-D-22-08991R1Nonalcoholic Fatty Liver Disease and Cognitive Impairment: a Prospective Cohort StudyPLOS ONE

Dear Dr. Cushman,

Thank you for submitting your manuscript to PLOS ONE. After careful consideration, we feel that it has merit but does not fully meet PLOS ONE’s publication criteria as it currently stands. Therefore, we invite you to submit a revised version of the manuscript that addresses the points raised during the review process.

Relevant to the reviews included, the authors should comment on the strengths and limitations of the study design, addressed questions related to the approach and resolve minor presentation errors. 

We look forward to receiving your revised manuscript.

Kind regards,

Nicholette D. Palmer, Ph.D.

Academic Editor

PLOS ONE

Journal Requirements:

Reviewers' comments:

Reviewer's Responses to Questions

**Comments to the Author**

1. If the authors have adequately addressed your comments raised in a previous round of review and you feel that this manuscript is now acceptable for publication, you may indicate that here to bypass the “Comments to the Author” section, enter your conflict of interest statement in the “Confidential to Editor” section, and submit your "Accept" recommendation.

Reviewer #2: All comments have been addressed

Reviewer #3: (No Response)

2. Is the manuscript technically sound, and do the data support the conclusions?

Reviewer #2: Yes

Reviewer #3: Partly

3. Has the statistical analysis been performed appropriately and rigorously? 

Reviewer #2: Yes

Reviewer #3: Yes

4. Have the authors made all data underlying the findings in their manuscript fully available?

Reviewer #2: Yes

Reviewer #3: No

5. Is the manuscript presented in an intelligible fashion and written in standard English?

Reviewer #2: Yes

Reviewer #3: Yes

6. Review Comments to the Author

Reviewer #2: (No Response)

Reviewer #3: The study by Cushman, et al. investigated the relationship the associations of NAFLD with risk of cognitive impairment using data from a prospective cohort study. The strengths of the study are the relatively large sample size and prospective cohort design. However, I have several concerns regarding study design and presentation of the data.

Major comments:

1. My major concern is that “NAFLD” is defined based on FLI, a non-invasive index calculated from waist circumference, body mass index, levels of triglycerides and GGT, and not based on validated imaging measures or a clinical diagnosis. Thus the reported association likely reflects the effect of metabolic syndrome (obesity and dyslipidemia) on cognitive decline, rather than the effect of NAFLD per se. The fact that the authors found no similar association of liver function tests (ALT, AST) with cognitive impairment underscores the problem.

While FLI has been shown to have reasonable accuracy in discriminating between NAFLD and non-NAFLD in some settings, I think it reflects a metabolic risk profile rather than disease per se.

2. Definition of cognitive decline (p. 6, lines 122-123): the authors state incident cognitive impairment was defined as impaired scores on at least two of the three follow-up tests, which were performed “every two years during follow-up” (lines 118-119). While cases and controls had similar mean age at baseline (Table 2), was there any difference in the length of follow-up between the two groups or the age at diagnosis/last follow-up? This information needs to be provided in Table 2 or the Methods.

3. Statistical analyses: can the authors provide more information about the weights used in the analysis?

4. Tables: please add p-values for comparison of the groups (of at least standardized mean differences). While some journals do not require p-values when presenting baseline characteristics, it is very difficult to detect important differences without scrutinizing the entire table and doing some back-of-the-envelope calculations. For example, the authors state that the groups with and without NAFLD (Table 1) did not differ by race; however, the percentages of Black subjects in the two groups were 39 and 29, which is quite a large difference.

5. Tables – categorical characteristics should be presented as number (%), not only %.

Minor comments:

6. Page 4, lines 67-68: “imply considered, toxin accumulation might cause neuronal damage, as might lowered production of protective substances.” Fragment/incomplete sentence. Please revise.

7. Page 10, line 208: “group classified with NAFLD during follow up”. I am confused. I thought the authors stated that NAFLD was only assessed at baseline, not follow-up.

7. PLOS authors have the option to publish the peer review history of their article (what does this mean?). If published, this will include your full peer review and any attached files.

Reviewer #2: No

Reviewer #3: No

---

## [Author Response · Author response to Decision Letter 1]

28 Jan 2023

I uploaded a file for this.

Response to Reviews Cushman et. al., PONE-D-22-08991R1

From the editors

The authors should comment on the strengths and limitations of the study design, addressed questions related to the approach and resolve minor presentation errors

Response: Strengths were already reviewed on page 19, and limitations extensively discussed on pages 19-20 (and we have added to these based on a reviewer comment). We have responded to each reviewer concern and resolved minor presentation errors.

Journal Requirements:

Response: References were checked. We did not cite retracted papers to our knowledge and did not add new references.

Reviewer Comments to the Author

Reviewer #3: The study by Cushman, et al. investigated the relationship the associations of NAFLD with risk of cognitive impairment using data from a prospective cohort study. The strengths of the study are the relatively large sample size and prospective cohort design. However, I have several concerns regarding study design and presentation of the data.

Major comments:

1. My major concern is that “NAFLD” is defined based on FLI, a non-invasive index calculated from waist circumference, body mass index, levels of triglycerides and GGT, and not based on validated imaging measures or a clinical diagnosis. Thus the reported association likely reflects the effect of metabolic syndrome (obesity and dyslipidemia) on cognitive decline, rather than the effect of NAFLD per se. The fact that the authors found no similar association of liver function tests (ALT, AST) with cognitive impairment underscores the problem. While FLI has been shown to have reasonable accuracy in discriminating between NAFLD and non-NAFLD in some settings, I think it reflects a metabolic risk profile rather than disease per se.

Response: We acknowledge the great point of the new reviewer and also agree that FLI is reasonably validated. We had included related comments about classification of NAFLD in the previous submission. We agree that metabolic syndrome and NAFLD will frequently co-occur and it may be impossible to tease apart in terms of what the causal pathway might be, even if we had a gold standard measure of NAFLD; the issue would be the same. In addition, although the FLI has components of metabolic syndrome in its formula, GGT, a biomarker for selected liver disorders, is not a component of metabolic syndrome. We adjusted for metabolic factors in the multivariable analysis, but this doesn’t mean there isn’t residual confounding. We also want to mention that individual measures of liver function do not adequately reflect NAFLD, so lack of associations for individual aminotransferase levels does not mean that the findings on FLI are invalid. For example, serum aminotransferase levels, which are often elevated in liver injury, are not specific biomarkers for NAFLD, and their levels can be normal in up to 25% of individuals with NAFLD (PubMed ID 31937252). Finally, AST/ALT >2 was indeed related to risk of cognitive impairment. 

We have added mention of the possibility of residual confounding by metabolic syndrome and the close connection of NAFLD with metabolic syndrome in the limitations section of the paper, page 19, line 370 in the tracked doc, “FLI presence may also relate closely to metabolic syndrome, and as such it would be difficult to separate the impacts of NAFLD and metabolic syndrome on risk of cognitive impairment. While we adjusted for factors involved in metabolic syndrome, we cannot say definitively that NAFLD is causal for the observed associations.”

2. Definition of cognitive decline (p. 6, lines 122-123): the authors state incident cognitive impairment was defined as impaired scores on at least two of the three follow-up tests, which were performed “every two years during follow-up” (lines 118-119). While cases and controls had similar mean age at baseline (Table 2), was there any difference in the length of follow-up between the two groups or the age at diagnosis/last follow-up? This information needs to be provided in Table 2 or the Methods.

Response: Test scores comprising the case definition for cognitive impairment were normed for age, which is why cases and controls have similar age. We selected the case and control groups based on data available in 2011 for follow up to their most recent cognitive test battery prior to this date (mean follow up 3.4 years). Any difference between cases and controls for follow up time would not seem relevant in this time span as all included participants had to have follow up cognitive assessment and these were done 2 years apart. We are unclear why age at last follow up would inform interpretation of the results. Based on the study design that age would be similar in the two groups.

3. Statistical analyses: can the authors provide more information about the weights used in the analysis?

Response: We added text on page 9 of the tracked doc at the bottom (line 186) to clarify this: “Baseline participant characteristics by NAFLD status and case control status were tabulated with weighting based on age group, sex, race to account for the stratified control sample selection so that the characteristics would reflect those in the larger REGARDS cohort.”

4. Tables: please add p-values for comparison of the groups (of at least standardized mean differences). While some journals do not require p-values when presenting baseline characteristics, it is very difficult to detect important differences without scrutinizing the entire table and doing some back-of-the-envelope calculations. For example, the authors state that the groups with and without NAFLD (Table 1) did not differ by race; however, the percentages of Black subjects in the two groups were 39 and 29, which is quite a large difference.

Response: We appreciate the reviewer pointing out the difference in race between those with and without NAFLD. It was an error in the text for us to say these prevalences were similar and we have edited this. We prefer to provide descriptive text only and avoid the use of p-values in the baseline characteristics table for several reasons. First, we are not testing hypotheses here but just describing the data. Second, statistical inference is not a valid method for determining the possibility of confounding in later analyses and may provide misleading information regarding this question. Third, statistical inference measures in a baseline characteristics table address the likelihood that a study exposure and a specific covariate are related in the underlying population; this question can be tangential to the study hypotheses and potentially distracting. Fourth, the use of statistical inference measures in a baseline characteristics table involves multiple comparisons and is likely to generate false positive associations. 

5. Tables – categorical characteristics should be presented as number (%), not only %.

Response: We elected to show percents since the data is weighted to the full cohort so there would then be 2 numbers and 2 percentages in each cell (one for the nested sample and one for the weighted values reflecting the full cohort). We hope the reviewer can agree this is simpler and preferable.

Minor comments:

6. Page 4, lines 67-68: “imply considered, toxin accumulation might cause neuronal damage, as might lowered production of protective substances.” Fragment/incomplete sentence. Please revise.

Response: apologies for this! We revised to, “…. pathways of an association are unclear. Simply considered, toxin accumulation …..”

7. Page 10, line 208: “group classified with NAFLD during follow up”. I am confused. I thought the authors stated that NAFLD was only assessed at baseline, not follow-up.

Response: This is correct. The phrase, “during follow up” was removed.

---

## [Decision Letter · Decision Letter 2]

20 Feb 2023

Nonalcoholic Fatty Liver Disease and Cognitive Impairment: a Prospective Cohort Study

PONE-D-22-08991R2

Dear Dr. Cushman,

We’re pleased to inform you that your manuscript has been judged scientifically suitable for publication and will be formally accepted for publication once it meets all outstanding technical requirements.

Kind regards,

Nicholette D. Palmer, Ph.D.

Academic Editor

PLOS ONE

Additional Editor Comments (optional):

Reviewers' comments:

Reviewer's Responses to Questions

**Comments to the Author**

1. If the authors have adequately addressed your comments raised in a previous round of review and you feel that this manuscript is now acceptable for publication, you may indicate that here to bypass the “Comments to the Author” section, enter your conflict of interest statement in the “Confidential to Editor” section, and submit your "Accept" recommendation.

Reviewer #3: All comments have been addressed

2. Is the manuscript technically sound, and do the data support the conclusions?

Reviewer #3: Partly

3. Has the statistical analysis been performed appropriately and rigorously? 

Reviewer #3: Yes

4. Have the authors made all data underlying the findings in their manuscript fully available?

Reviewer #3: Yes

5. Is the manuscript presented in an intelligible fashion and written in standard English?

Reviewer #3: Yes

6. Review Comments to the Author

Reviewer #3: (No Response)

7. PLOS authors have the option to publish the peer review history of their article (what does this mean?). If published, this will include your full peer review and any attached files.

Reviewer #3: No

---

## [Editor Report · Acceptance letter]

24 Feb 2023

PONE-D-22-08991R2 

Nonalcoholic Fatty Liver Disease and Cognitive Impairment:a Prospective Cohort Study 

Dear Dr. Cushman:

I'm pleased to inform you that your manuscript has been deemed suitable for publication in PLOS ONE. Congratulations! Your manuscript is now with our production department. 

Kind regards, 

on behalf of

Dr. Nicholette D. Palmer 

Academic Editor

PLOS ONE